# Patient perspectives on the usability and content validity of the assessment of burden of chronic conditions tool for post-COVID in the Netherlands: a qualitative study

Valerie H J Debie ⬤ , Loraine H L Peters ⬤ , Onno C P van Schayck, Jako S Burgers, Ramon P G Ottenheijm, Annerika H M Gidding-Slok

Care and Public Health Research Institute (CAPHRI), Department of Family Medicine, Maastricht University, Maastricht, Netherlands

**Correspondence to**
Valerie H J Debie;
v.debie@maastrichtuniversity.nl

## ABSTRACT

**Background** Post-COVID syndrome manifests with a diverse array of symptoms for which no standard care plan currently exists. Many questions were raised by patients, which underscored the need for a validated patient-reported outcome measure (PROM). Therefore, a post-COVID module was developed to be included in the Assessment of Burden of Chronic Conditions (ABCC-) tool. The ABCC-tool evaluates and visualises the perceived physical, emotional and social burden of one or multiple chronic disease(s) using a balloon diagram and aims to facilitate person-centred care and structured discussions between patients and healthcare professionals. This study explores the patients' perspective on the content of the ABCC-tool for post-COVID and the tool's usability in a home-based setting.

**Methods** All patients who completed the ABCC-tool for post-COVID were invited for an online semi-structured interview. We selected post-COVID patients who had used the tool in the past three months. Interviews were audio recorded and analysed using a thematic approach with Atlas.ti version 23.

**Results** Nineteen post-COVID patients (10 males, mean age 56) were interviewed between May and August 2024. The tool was regarded as user-friendly, and patients indicated they would use the tool again in the future. Patients valued the tool's broad range of topics, some of which are often overlooked in standard healthcare consultations. The tool was comprehensible and relevant according to all patients. The balloon diagram was easy to understand, but a legend explaining the colours of the balloons was preferred. Other suggestions for improvement included adding open-text fields and periodic reminders to increase usability and adding long-term data.

**Conclusions** The ABCC-tool is a promising instrument for post-COVID patients, offering a structured way to monitor and communicate experienced burden in addition to standard healthcare consultations. Refinements addressing usability and comprehensiveness are recommended to facilitate its integration into clinical practices.

---

### STRENGTHS AND LIMITATIONS OF THIS STUDY

⇒ The qualitative design provided in-depth information about the content and patients' perspective of the Assessment of Burden Chronic Conditions (ABCC-) tool.
⇒ Data were analysed using inductive and deductive approaches to increase the study's credibility.
⇒ Due to the lack of a clinical test for post-COVID, it was not possible to verify the patient's diagnosis.

---

## INTRODUCTION

Most people infected with SARS-CoV-2 (COVID-19) fully recover after the acute phase of the disease, regardless of the severity of their infection. However, according to the World Health Organisation (WHO), 6% experience persistent symptoms that hinder their daily life activities and decrease quality of life (QoL).[1–3] Over 200 different symptoms have been reported, manifesting in various combinations. These may include, but are not limited to, respiratory, cognitive and neurological symptoms, (chronic) fatigue, emotional burden and work impairment.[1 2 4] This condition, known as post-COVID or long-COVID, is defined by the WHO as the emergence and persistence of these symptoms for at least 2 months, occurring 3 months after the initial infection, without any other identifiable cause.[2]

Post-COVID patients reported persistent symptoms which raise many health-related questions, prompting them to visit their general practitioner (GP). Managing post-COVID is challenging due to the heterogeneity of symptoms and limited knowledge. To address these challenges, it is crucial that GPs take a comprehensive, holistic approach to patient care. This includes managing physical

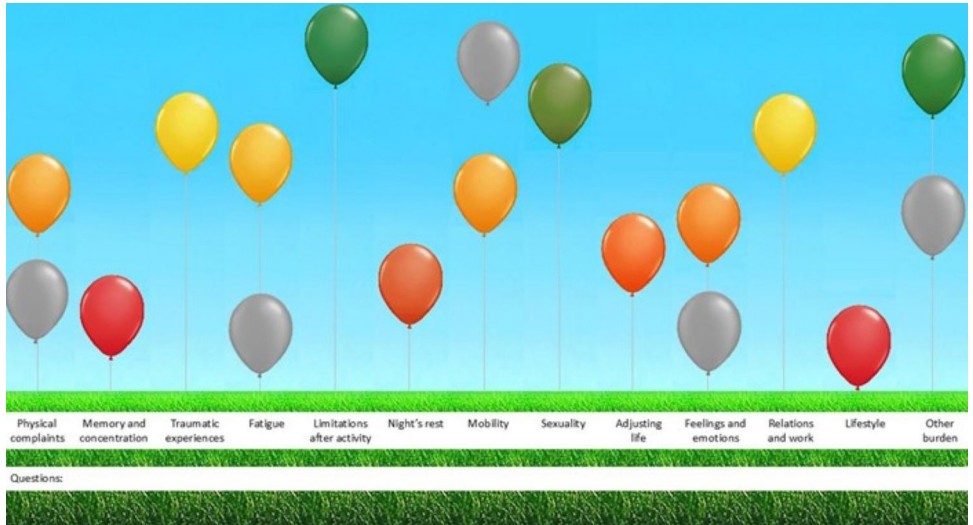

**Figure 1** The balloon diagram of the ABCC-tool for post-COVID. This diagram presents the results of the questionnaire from the ABCC-tool. Red balloons indicate a high burden of disease, while green balloons represent a low burden. Yellow and orange balloons depict intermediate scores, and grey balloons show the scores from the previous completion of the ABCC-tool. ABCC, Assessment of Burden Chronic Conditions.

symptoms, supporting mental well-being and social participation, and monitoring recovery. Additionally, GPs could play a role in referring patients to specialists or rehabilitation programmes when necessary. However, the absence of evidence-based guidelines and the heterogeneous nature of symptoms make this task particularly demanding, and a supporting tool is deemed necessary.[5–7]

Given the need for structured support, Lung Foundation Netherlands (an online Dutch forum and helpdesk for lung-related questions) decided in 2020 to develop a COVID-19-oriented tool to help patients with their concerns and coping with their symptoms. Due to the urgent need for patient support, it was decided to use already existing, validated and well-known patient reported outcome measures (PROMs) as a foundation. Given that acute COVID-19 primarily affects the lungs, PROMs focused on pulmonary symptoms were assessed.[8] Eventually, the Assessment of Burden of Chronic Obstructive Pulmonary Disease (ABC-) tool was selected and adjusted to create the Assessment of Burden of COVID-19 (ABCoV-) tool.[8] However, as the pandemic evolved, it became clear that a tool focusing solely on the acute phase of the disease provided only limited value to patients. Therefore, the ABCoV-tool was further adapted to serve as a post-COVID module within the Assessment of Burden of Chronic Conditions (ABCC-) tool.

As the successor to the ABC-tool, the ABCC-tool extends its focus beyond chronic obstructive pulmonary disease (COPD). Because of its modular format, every patient completes a set of generic questions, followed by one or more disease-specific modules. Until now, these disease-specific questions have been developed for asthma, COPD, type two diabetes mellitus, chronic heart failure, osteoarthritis, cardiovascular risk management and post-COVID.[9–11] The ABCC-tool was chosen because it measures burden of disease on a physical, emotional and social level, and because of its person-centred approach and stimulation of self-management and shared decision making. These are important factors for disease coping, improving health outcomes and reducing the use of healthcare services.[12] The tool includes a self-administered questionnaire with 41 questions addressing physical, emotional and social burden of disease, plus one open question, followed by a visual presentation of the results in the form of a balloon diagram (figure 1).[9 11] All questions are designed to be understandable at a B1 language level. This level indicates that patients can understand descriptions of events, feelings and wishes and is characterised by short and easy sentences.[13]

The ABCC-tool has been proven to increase perceived quality of care and patient activation.[14] A full description of the ABCC-tool and its intended use is published elsewhere.[9]

In this study, we aimed to assess the content validity of the ABCC-tool for post-COVID and to evaluate its usability from the patients' perspectives. To the best of our knowledge, no other tool has the capability to measure physical, emotional and social disease burden of post-COVID with the possibility to measure these disease burdens in other chronic conditions and visualise the results. Such tools are highly demanded by patients. Therefore, this evaluation was to ensure that the tool adequately covers all relevant content and to identify potential areas for improvement to make it easier for patients to use.

## METHODS
### Study design
This qualitative study used an interpretative phenomenological approach to explore the experiences of post-COVID patients with the ABCC-tool in the home setting. This method examines how individuals make sense of

their experiences with the tool and provides a structured framework for research design, data collection and analysis, enabling a detailed exploration of their perspectives.[15 16] The Consensus-based Standards for the Selection of Health Measurement Instruments (COSMIN) methodology for assessing the content validity of PROMs was used to assess the content validity.[17] The Consolidated Criteria for Reporting Qualitative Research (COREQ) checklist was used to ensure comprehensive and transparent reporting.[18] The full questionnaire is presented in online supplemental appendix 1.

Ethical approval was provided by the Medical Ethics Committee of the Maastricht University Medical Centre (METC 2022–3439). The study was funded by Pfizer and ZonMw; however, they did not have any involvement in the study design, data collection, analysis or manuscript writing process. All interviewed patients provided written informed consent before the interview.

## Study participants

All patients who had used the ABCC-tool online via CuraVista (www.curavista.health) platform were invited for an online interview via email by CuraVista. CuraVista is an online eHealth platform for managing chronic diseases through monitoring and self-management programmes. Patients can use this platform individually or together with their healthcare professional.[19] The aim was to interview 15 to 20 patients until data saturation was achieved. Patients were eligible if they had a self-reported diagnosis of post-COVID according to the WHO definition, completed the ABCC-tool within the past 3 months to minimise recall bias, and were proficient in the Dutch language. Before the start of the interviews, patients were specifically asked whether they could remember completing the questionnaire and the balloon diagram and whether they had post-COVID. Patients were excluded if they could not remember using the tool or seeing the balloon diagram. Patients were invited in small batches to account for potential non-responses and scheduling conflicts.

## Data collection

Data were collected through single individual semi-structured online interviews in Dutch, conducted between May and August 2024 by one female researcher (VHJD), who was trained in qualitative research. All other researchers involved had experience in performing qualitative research. Relevant topics for the interview were outlined in a topic guide developed by the research team, which was pilot tested during the first interview. This topic guide was iteratively adjusted between interviews to address emerging themes to enhance data richness, such as the order of the questions. The final topic list is provided in online supplemental appendix 2. The two main topics included the content validity of the ABCC-tool and its usability. Prior to the interviews, patients received a printed copy of the tool's questionnaire and the balloon diagram by mail to recall the tool and facilitate discussion. All interviews were audio-recorded with patients' consent. No field notes were made during the interviews. The interviewer had no relationship with any of the patients. Patients were only familiar with the aim of the research. The interviewer was introduced to the participants as a health scientist and researcher at Maastricht University.

## Data analysis

Audio recordings were subsequently transcribed using a custom implementation of WhisperX, which provides audio transcription based on Whisper (large-v2 model) and speaker diarisation. This transcription process was performed locally on Maastricht University infrastructure/hardware.[20] Transcripts were 100% checked for correctness. Data analysis was conducted using a thematic approach with Atlas.ti version 23, software for qualitative data analysis. To analyse the transcripts, we used inductive — the process to identify new themes from textual data – and deductive – the process of applying pre-established codes to the transcripts — coding.[21] New themes were systematically monitored in between interviews to observe for data saturation and information power. Related codes were grouped into broader categories, and overarching themes were synthesised to gain a complete understanding of the patients' perspectives.

This iterative thematic analysis was reviewed and discussed with all members of the research team until consensus was reached. Initially, two researchers (VHJD and LHLP) independently coded 100% and 50% of all transcripts, respectively, to identify initial codes. The coding was compared, and discrepancies were resolved through discussion, enhancing inter-rater reliability.

## Patient and public involvement

No patients or public representatives were involved in any part of this study.

## RESULTS

In total, 5615 patients were invited for an interview, of which 27 interviews were started and 19 were completed and included in the analysis. figure 2 presents the process of selecting and including patients. Ten men and nine women participated with a mean age of 56. Furthermore, approximately half of the patients had comorbidities. Eleven patients had a high educational level. No patients with a low educational level were included. In table 1 the details per patient are presented. The length of the interviews ranged from 20 to 56 min. Data saturation was achieved for all topics, except comprehensiveness, after 16 interviews. In no interview were additional people present. One interview (interview 11) was cut short early due to concentration issues.

## General evaluation of the ABCC-tool

Patients appreciated the tool's questionnaire and balloon diagram for providing a clear overview of both their

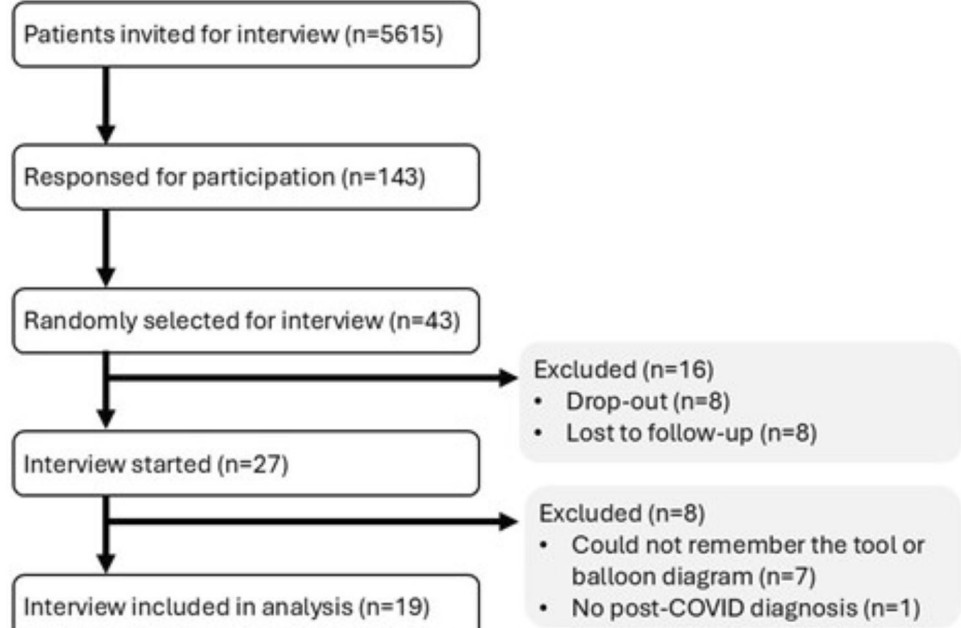

**Figure 2** flowchart of the inclusion of patients for an interview.

current and past health status. The diversity of questions was also highly valued, especially those topics that were not typically linked to post-COVID by their healthcare professionals.

Patient 12: "I think it's a good tool. I believe it provides a lot of people with valuable insights into certain complaints, which might be related after all, things you wouldn't always immediately link, but that could indeed be connected in some way."

Several patients found it challenging to complete a questionnaire about their health over the past 2 weeks. They noted having both good and bad days, which made it difficult to answer the questions consistently. Some patients also struggled to remember their health status over the previous 2 weeks or from their last use of the tool. To address these challenges, some patients suggested incorporating extra text boxes to allow for personal notes.

Patient 13: "And what might be helpful […] is if there were spaces to write something down about your situation at that moment. […] For example, if I look at my balloons now, I think, yeah, I have no idea what I was doing then, where I was, or if I was doing different things."

Most patients had used the ABCC-tool multiple times or expressed a willingness to use it more frequently. Nonetheless, many reported forgetting about the tool and suggested receiving periodic reminders, such as monthly or quarterly emails. Others mentioned they would only consider using the tool again if their symptoms changed.

Patient 11: "I think if I received an email saying it's been three months since you filled in the ABCC-tool.

Click here to fill it in again, I would be more inclined to do it."

Seven patients required a more detailed explanation of how the tool worked or what to do with it.

Patient 3: "I see the balloons, but I'm not really sure what I'm supposed to do with them."

### Usability of the ABCC-tool

Not all patients could recall exactly where they found the ABCC-tool, but most patients had come across the ABCC-tool during online searches aimed at determining whether their symptoms and disease burden aligned with post-COVID. Others were alerted through Facebook support groups, an email from C-support (a foundation of the Dutch Ministry of Health, Welfare and Sport that supports and advises people with post-COVID), or by chance through a Google search for treatments.

Patient 18: "C-support, I believe. I get a lot from them because once you're connected with them and that sort of things, you receive all kinds of information, so to speak."

Nobody experienced any difficulties with the practical use of the ABCC-tool. The website and login process did not present any problems either.

Patient 6: "No, I found it [the website] quite easy."

The aim of the ABCC-tool was not clear to everyone. Some patients thought it was part of a larger study to find an effective treatment for post-COVID, and that their participation would contribute to the development of new treatments or pharmaceutical interventions. However, most patients understood the aim of the tool.

 Debie VHJ, *et al. BMJ Open* 2025;**15**:e109201. doi:10.1136/bmjopen-2025-109201

**Table 1** Characteristics of the patients

| Patient nr. | Age * | Gender | Other chronic conditions | Educational level† |
|---|---|---|---|---|
| 1 | Mid 50s | Female | Apnoea<br>Lymphedema<br>Thyroid disorder | Medium |
| 2 | Late 50s | Female | – | High |
| 3 | Early 60s | Female | Asthma<br>Eczema | Medium |
| 4 | Early 50s | Female | Asthma | High |
| 5 | Early 60s | Male | Musculoskeletal disorder<br>Thyroid disorder | High |
| 6 | Mid 50s | Female | – | Medium |
| 7 | Late 50s | Female | – | High |
| 8 | Late 70s | Male | Apnoea<br>Cardiovascular disease<br>Rheumatic disease | High |
| 9 | Early 50s | Male | Asthma | Medium |
| 10 | Early 50s | Male | – | Medium |
| 11 | Early 40s | Male | – | Medium |
| 12 | Mid 60s | Male | – | Medium |
| 13 | Late 50s | Female | – | High |
| 14 | Mid 50s | Male | COPD | Medium |
| 15 | Late 40s | Male | – | High |
| 16 | Early 40s | Female | – | High |
| 17 | Early 50s | Male | – | High |
| 18 | Late 50s | Male | Rheumatic disease | High |
| 19 | Mid 60s | Female | – | High |

*Age categories: early: age ending on 0-1-2-3, middle: 4-5-6, late: 7-8-9
† Dutch educational levels: *Low* = primary school and preparatory vocational secondary education (VMBO); *Medium* = senior general secondary education (HAVO), university preparatory education (VWO) and secondary vocational education and training (MBO); *High* = higher professional education (HBO) and research-oriented education (WO)[42].
COPD, chronic obstructive pulmonary disease.

Patient 13: "I thought it was also for a broader study. I didn't realize it was just for myself and possibly a healthcare provider. I actually didn't know that."

Most patients expressed a willingness to recommend the tool to other post-COVID patients, although some indicated they did not know any other post-COVID patients. One patient mentioned they would only recommend the tool if additional medical support would follow afterwards. Furthermore, two patients expressed a preference for more medical support after completing the ABCC-tool.

Patient 17: "It needs to be monitored. But if someone is consistently red month after month, then something is really wrong. And at that point, someone should be guided, like, hey, I think you need to be doing something else than what you're doing now. […] Or maybe referred to someone else."

Almost all patients found the questionnaire manageable in terms of length, as it could be paused and resumed

at any time. However, one patient mentioned a lack of concentration to complete it and found the questionnaire too long.

Patient 7: "Yes, it was a nice experience. And I definitely liked that it can be saved in between. So, you don't have to complete it all at once. I did it in two parts."

### Content of the ABCC-tool

The questions in the questionnaire were clear to all interviewed patients.

Patient 2: "Otherwise, I find it very clear. The language is very clear, and also how you rate [the answer from 0-6] it, so to speak, or the choices you have. Yes, it's just very clear to me."

Regarding the comprehensiveness of the tool's questionnaire, data saturation was not achieved during the interviews. However, patients agreed that the most

burdensome and most frequent burdens are included in the questionnaire. Additionally, four patients suggested including open-text fields throughout the questionnaire, allowing respondents to add personal notes about their current health status. This would facilitate easier comparison of responses across assessments. Another patient proposed incorporating a multiple-choice question to document any changes in areas such as health status, work, family situation or other relevant personal factors.

Additionally, patients recommended adding questions regarding the type of therapy undergone, general pain experiences, balance difficulties, weight fluctuations due to oedema, limitations in daily routines, symptom duration, feelings of being understood by government agencies and restrictions experienced after intense physical activity.

Several suggestions were made to split questions that contain multiple items into separate questions. For example, two patients recommended breaking down the question on low, moderate and vigorous physical activities into smaller steps to enable the possibility of more green balloons.

All questions were deemed relevant, except for one regarding sexuality, which was considered inappropriate by some in case of sharing the balloon diagram with their employer. Another patient felt this question was irrelevant for older individuals.

Patient 18: "I can imagine that the topic of sexuality in such a chart… it's not very convenient if you want to show it to your boss. That's not something they should be involved in."

The answer options were well-received, with most patients finding both the wording and number of response choices suitable.

Patient 11: "I think 0 to 6 is good. Actually, very good."

### Visual representation

Most patients found the coloured balloons easy to understand and fun to use. Furthermore, the balloon diagram made it easier to provide the patient with helpful insights into their health status. However, some patients were confused about whether the height of the balloons (eg, low balloons indicating low burden) or the colours (eg, green balloon indicating low burden) indicated the burden of disease or were uncertain about the grey balloons and required further clarification. Those who had difficulty understanding the balloons suggested adding a legend or marking 'high' and 'low' burdens of disease on the vertical axis. Another patient expressed interest in seeing more long-term data, for example, results from the past six or seven assessments.

Patient 12: "At least putting something down like high-low or a lot or not. […] That would have been much clearer, yes."

All patients recognised their health status in the balloon representation. While some appreciated the simplicity of the balloons for providing a clear overview of their current situation, others found the abundance of red balloons confronting. Furthermore, patients varied in their level of acceptance regarding their situation, with some demonstrating greater acceptance than others.

Patient 2: "It was really confronting for me, but it was clear. You really see it there, that was very helpful for me, to really tell myself […] it's really real."

Some patients shared the balloon diagram with healthcare professionals to discuss their current and past health, although not all healthcare professionals expressed interest in the balloon diagram. Those who preferred to use the ABCC-tool and balloon diagram together with a healthcare professional would like to use it during visits with GPs, physical therapists, occupational therapists, rehabilitation specialists or company doctors. While not everyone considered discussing the diagram with a healthcare professional, some patients expressed an intention to do so in the future. Others shared the balloon diagram with family, friends or colleagues to explain their experiences with post-COVID. However, some patients felt they did not need the diagram, as they were able to explain their condition effectively without it. One patient found the diagram too confronting to share with others.

Patient 1: "No, I never showed the balloons. But I do know where my stumbling blocks are and what I need to work on."

Patient 6: "Yes, with my family members, my children. […] I find it nice to explain to them why I can't do certain things at times. Why I'm occasionally a bit down. […] They also understand it better."

### DISCUSSION

This study aimed to assess patients' perspectives on using the ABCC-tool for post-COVID, and the content validity of the questionnaire within the tool. As patients are experiential experts in their disease, in-depth interviews with them provided meaningful insights, emphasising essential elements for the tool's use. The ABCC-tool was regarded as user-friendly and provided a comprehensive overview of the post-COVID burden.

Findings from this study confirm that the tool includes the most commonly reported post-COVID burdens and offers patients the opportunity to document additional concerns through an open-text feature. Furthermore, patients appreciated the tool for multiple reasons, meaning that the tool appears useful. First, the diversity of topics included in the tool, particularly as many of these are not routinely addressed in their healthcare. Second, patients found the tool's language accessible, because of its short sentences and simple wording.[22] Third, the length of the questionnaire was acceptable. Fourth,

the visualisation gave an insightful overview of disease burden. Finally, the tool was easy to find and use online.

For a few patients, the aim of the tool, and this study, was unclear, and they thought that the tool was part of a larger study to find a therapy or cure for post-COVID. This highlights how easily PROMs can be misunderstood, with potential consequences for engagement and expectations. To prevent such misperceptions, it is important that future use of the ABCC-tool is accompanied by clear communication about its purpose as a self-management and shared decision-making instrument, rather than as a diagnostic or therapeutic tool.

Since the emergence of post-COVID, several PROMs have been developed and validated to assess the disease burden, functional status, symptom burden, QoL and post-COVID-related stigma.[23] Examples of tools are the Symptom Burden Questionnaire for Long COVID (SBQ-LC), the Post COVID-19 Condition Stigma Questionnaire (PCCSQ), and the Long COVID Symptoms and Severity Score (LC-SSS scale).[24–27] When comparing these tools to the ABCC-tool, we concluded that the ABCC-tool (44 items) is similar in length compared with the LC-SSS (44 items) and PCCSQ tool (40 items), with all three being scored on a 4 to 7 point Likert scale.[24–27] However, the ABCC-tool stands out, as none of these other tools specifically focus on the core dimensions of disease burden as defined by the WHO (ie, physical, emotional and social burden) and incorporate a visual representation of the results. It is important, however, to note that while these tools differ in scope and structure, no formal comparison was conducted regarding user experience or completion time. Therefore, although patients rated the ABCC-tool as user-friendly in this study, no conclusions can yet be drawn about its relative usability or efficiency compared with other available post-COVID PROMs. Future studies could explore these aspects in more depth through head-to-head comparisons.

In the current study, patients specifically preferred this visual representation. Although the balloon diagram was not immediately clear to everyone, a brief explanation made it fully understandable. Therefore, the inclusion of a legend for the balloon diagram was considered essential. This is congruent with recommendations that have been made in studies with the ABC-, ABCC-, and Assessment of Burden of ColoRectal Cancer (ABCRC)-tool, to facilitate better understanding of the diagram.[12 28 29] The balloons gave patients understandable insights into their health status, facilitating structured discussions with healthcare professionals. However, some patients had numerous red balloons which felt confronting and negatively impacted their motivation, for example, to adopt a healthier lifestyle. It is important to note that many of these patients had comorbidities, making it difficult to determine whether their reported burdens were due to post-COVID or these pre-existing conditions. Therefore, incorporating post-COVID into the ABCC-tool is more beneficial than developing a stand-alone tool, as the ABCC-tool includes multiple chronic conditions in

a single PROM by integrating the relevant modules for each condition.[9–11]

Additionally, other adjustments were recommended to make the tool more suitable for use. These include, among others, more open-text fields to accommodate personal notes. Given that memory loss and cognitive impairment are well-documented burdens of post-COVID,[2] patients often struggle to recall their experiences from previous weeks. This leads to difficulties in distinguishing day-to-day changes. Therefore, patients preferred the option to make notes to ensure more accurate responses. Furthermore, both patients in this study and those utilising the ABC-, ABCC- and ABCRC-tool have suggested the implementation of periodic email reminders to encourage more frequent completion of the questionnaire.[12 28 29] Research indicates that sending email reminders can significantly improve the completion rate of PROMs.[26] Patients indicated that the tool would be more relevant if they had more frequent follow-up results to better monitor their progress.

Only limited evidence is available on optimal treatment strategies for post-COVID, as it is a relatively new disease. Currently, most treatment programmes involve wait-and-see approaches, physical exercise and lifestyle recommendations.[30 31] Ideally, post-COVID care should move towards a more personalised and holistic approach. However, this is challenging due to the lack of robust evidence on what interventions are effective per symptom, the high number of reported symptoms and the substantial variability in burden among individuals.[2 5 6] Given these uncertainties, the ABCC-tool could play a role in addressing this gap.

**Strengths and limitations**

This study has several strengths and limitations. The qualitative study design provided in-depth results in the content validation and the perspective of patient of the ABCC-tool. The use of both inductive and deductive approaches strengthened the study's credibility. The inductive coding process minimised confirmation bias by allowing themes and patterns to naturally emerge from the data, rather than being predefined. Meanwhile, deductive analysis reduced researcher bias by cross-verifying themes with existing frameworks.[32–35]

Data saturation was not fully achieved on comprehensiveness; however, we systematically monitored whether new symptoms or themes emerged during the interviews. While some additional complaints were mentioned, these largely overlapped with issues already identified or were not applicable for the ABCC-tool. We therefore argue that information power was sufficient, as the sample provided rich and relevant data on the clarity and relevance of the questionnaire items. Factors such as the study's aim and the quality of the collected data play an important role in determining whether the sample holds sufficient information for meaningful analysis.[36] Due to the complexity of post-COVID and the aforementioned over 200 symptoms that are associated with this condition,[2] it is not feasible to include all symptoms in the questionnaire.

Furthermore, the included symptoms reflect the most prevalent burdens described in the literature on post-COVID, such as fatigue and cognitive dysfunction, which were consistently confirmed by our participants.[1 2 4] We therefore decided that this issue can only be solved by incorporating an open question asking about complaints other than those already mentioned. This way, the most prevalent burdens are included into the tool, with space for additional burdens to report.

In total, only 143 of the 5615 invited patients were eligible and interested in participating. We suspect that concentration problems commonly associated with post-COVID may have discouraged many patients from completing the interviews, as these were perceived as too demanding, among others. This may have led to selection bias: those who volunteer to participate in scientific research are mostly women, higher educated and score higher on cognitive tests.[37] The underrepresentation of patients with a low educational level could potentially limit the external validity of the findings.

We interviewed a roughly equal number of men (n=10) and women (n=9), including patients with and without comorbidities. However, we were not able to take into account educational level when including patients, as this information was unavailable beforehand. Despite the random selection, no participants with lower educational levels were included. Regardless, we expect the tool's questions to be accessible and readable for individuals with lower education due to their B1 language level. Furthermore, previous studies on the ABCC tool's content validity and patient experiences included participants with both higher and lower educational levels.[10 12] None of the patients reported difficulties with using the tool or understanding its language. Although the current study focused on a different patient population, the tool's purpose, functioning and linguistic level remain unchanged. For future research, we recommend collecting basic patient information to enable more targeted recruitment and selection.

Our inclusion criteria were not objectively verifiable. First, we could not confirm whether the patient had post-COVID or not, as there is no standardised or validated diagnostic method for this condition available. Currently, diagnoses are typically made by healthcare professionals or self-reported by patients, based on a set of symptoms. Second, it was impossible to verify the timing of patients' last use of the tool online, as these data were not accessible by the research team. To address these limitations, patients were asked at the start of the interview to specify who provided their post-COVID diagnosis and elaborate when this was self-diagnosed. They were also asked to indicate when they last used the ABCC-tool and whether they could remember this to minimise recall bias. If either of these two criteria was not met, the interview was terminated. For some patients, lack of concentration was a significant burden of post-COVID, which impacted their ability to participate in lengthy interviews. One interview had to be cut short on the demand of the

patient. Additionally, some patients mentioned at the beginning of the interview not to be able to talk longer than a specific time. In that case, the interviewer started with the most important topics of the study (i.e. usability of the tool and content validation of the questionnaire). For the same reason, transcripts were not returned to the patients because this would be too burdensome. However, we acknowledge that offering participants the option to review their transcripts could have provided an additional opportunity for validation and feedback, and we will consider this approach in future studies.

### Implications for future research and practice

A few adjustments (open text fields, periodic reminders, including a figure legend, and clarifying the purpose of the tool) need to be made to the tool, based on the findings in this research. Furthermore, although the questions of the tool have now been validated, a thorough evaluation of the other aspects of the tool is in place. For example, the recommended score calculation and lifestyle advice regarding physical activity might not be appropriate in the case of post-exertional malaise (PEM). This needs further investigation.

After these adjustments and further evaluation of the scores, a pilot study should be conducted to evaluate the usability and feasibility of the tool in healthcare settings. This study should include interviews with patients and healthcare professionals to evaluate the barriers and facilitators for using the tool and study the usability from different perspectives. Additionally, evidence-based treatment recommendations for each domain must be developed and integrated into the tool before clinical implementation. Technical aspects, such as seamless integration into electronic health records, should also be addressed to ensure its practical use in healthcare systems. These steps would enable the ABCC-tool to enhance care for post-COVID patients.

Given the positive results of the ABCC-tool in assessing disease burden in patients with post-COVID, further research is needed to explore its implementation in clinical practice. As post-COVID is a relatively new condition, the first step is to establish guidelines supporting person-centred care and shared decision making. The ABCC-tool could play a key role in supporting this by providing a structured and efficient holistic assessment of patients' burden and needs.

Although the ABCC-tool was originally developed in the Netherlands, future research could explore its implementation in other countries. This would require careful translation and cultural validation to ensure that the tool is both linguistically accurate and contextually appropriate for use in different healthcare settings. The first interest in the ABCC-tool of other countries has been reported already. Furthermore, it may be valuable to explore the potential application of the ABCC-tool for post-COVID in patients with myalgic encephalomyelitis/chronic fatigue syndrome (ME/CFS), as the burden associated with

ME/CFS closely resembles that of post-COVID. Both conditions are characterised by substantial health, economic and social impacts, which are often underestimated due to limited awareness and understanding of these diseases.[38] In addition to the similarities in perceived disease burden, several studies have examined the prevalence of post-COVID and ME/CFS. These studies reported that 8–58% of participants met criteria for both post-COVID and ME/CFS.[39–41]

## CONCLUSION

The ABCC-tool is seen as a useful tool to assess experienced burden of disease in patients with post-COVID and to monitor the course of symptoms in time. The most appreciated aspects of the tool are the balloon diagram, which presents the current and past health situation of the patient, and the diverse range of questions in the questionnaire. Furthermore, the content of the questionnaire is praised for being comprehensive, comprehensible and relevant for post-COVID. Based on the current and previous studies on the ABCC-tool, the tool for post-COVID holds significant potential to enhance person-centred care for post-COVID patients and is suitable for integration into clinical practice.

**Acknowledgements** We would like to thank all the patients who showed interest in participating in this study. Furthermore, we would like to thank CuraVista for their contribution to the development and implementation of the ABCC-tool. In addition, this research was made possible, in part, using the Data Science Research Infrastructure (DSRI) hosted at Maastricht University.

**Collaborators** Not applicable.

**Contributors** VHJD, RPGO, and AHMG-S were involved in the study design. VHJD conducted the interviews. VHJD and LHLP performed the coding. All the authors reviewed the thematic analysis and reached a consensus. Furthermore, all the authors contributed to the writing process, provided feedback, and approved the final version of the manuscript. AHMG-S is the guarantor.

**Funding** This study was funded by ZonMw (10430302130003) and Pfizer (75242181). The ZonMw grant was received by Jean WM Muris, with JSB as project leader. The grant of Pfizer was received by AHMG-S. More information is available on https://www.pfizer.nl/. The funders had no role in study design, data collection and analysis, decision to publish or preparation of the manuscript.

**Competing interests** All other authors have no competing interest to declare.

**Patient and public involvement** Patients and/or the public were not involved in the design, conduct, reporting or dissemination plans of this research.

**Patient consent for publication** Consent obtained directly from patient(s)

**Ethics approval** Ethical approval was provided by the Medical Ethics Committee of the Maastricht University Medical Centre (METC 2022-3439). All interviewed patients provided written informed consent before the interview.

**Provenance and peer review** Not commissioned; externally peer reviewed.

**Data availability statement** Data will be stored in a repository called DataHub. Data will be available upon reasonable request.

**ORCID iDs**
Valerie H J Debie https://orcid.org/0009-0004-1448-2837
Loraine H L Peters https://orcid.org/0009-0009-6712-3180

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
