## [Reviewer comments · BMJ Open]

ARTICLE DETAILS

Title (Provisional)

Patient perspectives on the usability and content validity of the Assessment of Burden of Chronic Conditions (ABCC-) tool for post-COVID in the Netherlands: a qualitative study

Authors

Debie, Valerie HJ; Peters, Loraine H L; van Schayck, Onno CP; Burgers, Jako S.; Ottenheijm, Ramon PG; Gidding-Slok, Annerika HM

VERSION 1 - REVIEW

Reviewer	1
Name	GUIDO, GIACOMO
Affiliation	University of Bari Aldo Moro
Date	28-Aug-2025
COI	None

Your manuscript addresses an important and underexplored aspect of post-COVID management, namely the usability and content validity of a PROM tailored to this complex condition. The qualitative approach provides valuable patient-centered insights, and your discussion situates the ABCC-tool well among other instruments. However, several issues need further attention. Most notably, the lack of participants with lower educational levels and the limited number of respondents relative to invitations cast doubt on representativeness. Moreover, patients' misunderstanding of the tool's purpose raises concerns about communication and implementation. Expanding your reflections on these points, clarifying recruitment challenges, and strengthening future directions (integration into EHRs, pilot testing in clinical practice, inclusion of diverse patient groups) would enhance the manuscript considerably. Overall, this is a promising study that, with some refinements, could make a meaningful contribution to patient-centered post-COVID care.

- The study acknowledges that participants predominantly came from medium to high educational backgrounds, with no representation of individuals from low educational groups. This limitation deserves deeper reflection because educational level can directly influence health literacy, digital literacy, and therefore usability of patient-reported outcome tools. Patients with lower education or lower health literacy may find the ABCC-tool more difficult to navigate, potentially reducing its real-world applicability. I would suggest adding a section that explicitly discusses how this exclusion may have shaped the findings and what steps could be taken in future studies to ensure more equitable inclusion (for example, targeted recruitment or oversampling strategies for underrepresented groups).

- At the same time, and this is connected to the previous comment in a way, the extremely high attrition rate is concerning: out of more than 5,000 invited patients, only 19 interviews were completed. Even if this was due to technical, logistic, or patient-related reasons, such a discrepancy raises questions about the representativeness of the final sample. It would strengthen the paper to discuss potential reasons for this attrition (e.g., lack of interest, digital barriers, survey fatigue, recruitment strategy) and to consider how these factors might limit external validity. For example, patients who declined participation may systematically differ from those who took part (e.g., in age, severity of post-COVID symptoms, or socio-economic status).
- Some participants misunderstood the purpose of the tool, believing it to be related to treatment trials or clinical interventions rather than a self-assessment instrument. This is a crucial finding because it suggests that communication around PROMs can strongly influence patient engagement and expectations. The discussion would benefit from an expanded reflection on how such misperceptions can be prevented
- Although the funding sources (Pfizer, ZonMw) are declared, the manuscript could be strengthened by clarifying the extent of funders' involvement in the study. For example, readers should be reassured that funders were not involved in study design, data collection, analysis, interpretation, or manuscript writing. Providing this explicit reassurance would help mitigate any concerns about potential conflicts of interest, particularly given that tool validation studies may lead to broader adoption and scaling
- Several patient quotations are very long, which affects readability. Consider condensing them while retaining their meaning.
- "ENGLISCH VERSION" in the appendix should be corrected to "ENGLISH VERSION."
- Ensure uniform use of terms such as ABCC-tool, PROM, GP, and CXR throughout the manuscript. Currently, some are written with hyphens and others without
- In the conclusion, "balloon scheme" is used, while earlier sections use "balloon diagram", better to harmonize
- The paper briefly notes directions for future research, but this could be expanded. For example, it would be valuable to suggest mixed-methods studies that include both patient and clinician perspectives to capture usability from both sides of the consultation. Plus, could it be implemented in other health systems besides the dutch one? I would explore transferability.
- The introduction provides a good overview of PROMs in post-COVID, but it could be strengthened by acknowledging that post-COVID outcomes extend beyond symptom monitoring to include functional and socio-economic consequences, such as impaired working ability. Recent studies have documented long-term work impairment up to two years after COVID-19 hospitalization (<https://doi.org/10.3390/v16050688>), which underscores the importance of PROMs that can also capture the impact of post-COVID on patients' ability to participate in daily and professional life

Affiliation **Henry M Jackson Foundation for the Advancement of Military Medicine Inc, IDCRP**

Date **09-Sep-2025**

COI **None**

Overall: The manuscript is qualitative review of user feedback rather than a study. The article itself lacked clarity of what exactly is being assessed and the justification for doing so. Substantial revision is needed to better frame the contribution of this work.

The tool being assessed does not appear to be very useful, and the goal of evaluating the tool is not fully justified.

Reviewer **3**

Name **Blome, Christine**

Affiliation **University Medical Center Hamburg-Eppendorf , Health Services Research in Dermatology**

Date **15-Sep-2025**

COI **None**

Thank you for the opportunity to review this interesting paper. It presents the findings of the qualitative content validation of a tool designed to assess the patient burden from Post-COVID (PC), called the ABCC-tool. It consists of 42 items and a graphical presentation of the individual results. The tool is intended to support PC care, which is important given the inadequate care available to these patients in many countries. Testing for content validity, i.e., testing whether the tool actually measures what it is intended to measure, is a crucial step in PROM development that is frequently neglected in research.

The manuscript is easy to read and clearly structured, and most of the relevant information is provided (but see below). It is very helpful that the authors explicitly specify the purpose of the tool and the context to be used in, i.e., health care (as opposed to, for example, clinical trials).

I have a number of comments and suggestions about the paper, particularly concerning its conclusions about the tool and the populations to which the results can be generalized.

Abstract:

If possible given the limited word count, it might be helpful to specify the context in which the patients completed the tool already in the abstract.

Introduction:

There is contradictory information about the exact construct to be measured. For example, "offering a structured way to monitor and communicate symptoms" seems to indicate that the construct to be measured is PC symptoms, while elsewhere the construct is referred to as "physical, emotional, and social burden of disease".

I believe that the figure of "ten to twenty percent experience persistent symptoms that hinder their daily life activities, and decrease quality of life" is probably outdated and was only true

only in the early years of the pandemic. I suggest referencing more recent studies on incidence and/or prevalence.

The considerable overlap between PC and ME/CFS could be mentioned, as could the phenomenon of post-exertional malaise (PEM), which is reported even more frequently.

Methods:

Patients obviously provided an email address when completing the tool online. Was this mandatory? Did they agree to be contacted for study invitation?

As the COSMIN guidelines provide recommendations on how to conduct content validation studies, you may wish to refer to these and which of these recommendations have been met in your work.

"This qualitative study used an interpretative phenomenological approach"; elsewhere, you state that it used a thematic approach. I am not sure whether this work actually qualifies as interpretative phenomenological; from the methods and results, it seems to me more like a thematic analysis / content analysis.

Do I understand correctly that the ABCC-tool, which had been developed for use in other chronic diseases, has not been adapted for use in patients with PC, meaning that this is the first study to investigate whether its content is appropriate to PC? I wondered because the item "did you suffer from a strange taste or smell, or tasted/smelled nothing" seems to be quite PC-specific. On the other hand, it is usually recommended that the development of a tool is performed with patient input (via qualitative data). This may explain why some potentially quite relevant impairments are not covered, such as PEM (while feeling worse after exercising is covered, feeling worse after any kind of activity is not).

It is all the more important to thoroughly test the appropriateness and completeness of the tool with this new patient group. However, if I understand correctly, the tool was not presented again in the interviews; instead, participants had to rely on their recollection of the items. This severely limits the conclusions that can be drawn regarding comprehensibility and in particular completeness of the tool's content, even more so as multiple patients had to be excluded as they did not remember using the tool at all. The COSMIN guidelines, for example, request that patients should be asked about each item separately, which has not been done here.

According to the interview guide, patients were asked whether they thought the ABCC tool was complete. However, they were not provided with any information about what the instrument intends to measure. What exactly does "complete" refer to in this context?

Were the computer-assisted transcriptions checked for correctness?

How is lifestyle transformed into balloon colour? For example, "moderately intense physical exercise for 30 minutes" (item 36) may be considered an unhealthy lifestyle choice for many people with PEM; therefore, a negative response should not be translated into a red balloon.

Results:

How did you assess which other chronic conditions the participants had?

"One interview had to be cut short on the demand of the patient. Additionally, some patients mentioned at the beginning of the interview not to be able to talk longer than a specific time. In

that case, the interviewer started with the most important topics of the study". How many patients exactly had limited or missing data compared to the interview guide? This information could be added to the patient characteristics table.

You state that data saturation was not achieved for comprehensiveness but chose not to include additional patients. In the discussion, you argue that "Due to the complexity of post-COVID and the aforementioned over 200 symptoms that are associated with this condition, it is not feasible to include all symptoms in the questionnaire. We therefore decided that this issue can only be solved by incorporating an open question" – I believe that this may be insufficient. More information is needed on how data saturation has been assessed and on the empirical basis for the statement that "the most prevalent burdens are included into the tool".

I also have some concerns about the heterogeneity of the sample, and therefore the generalisability of the results to the overall population of people with PC. Out of 5615 patients invited for an interview, only 143 responded, which seems quite low. The youngest person included was 41, so the perspectives of younger patients are not represented. Also, severely affected participants, e.g. those with severe ME/CFS, were not (and probably could not be) included, which should be discussed.

Many issues with the tool were found during the interviews, but these did not lead to revisions of the tool (except for the addition of a legend). Examples of issues that were not addressed include the recall period, the purpose of the tool and how to interpret the graphic. Instead, it is recommended that only "Minor adjustments, e.g. open text fields and periodic reminders, need to be made to the tool, based on the findings in this research" (which should be specified: which adjustments exactly should be made, as only examples are provided?). It seems to me that the findings do not fully justify the conclusion that the tool showed content validity and feasibility.

Discussion:

The need for this new tool and why it is innovative as compared to existing tools are discussed. Note that there are reviews of existing PROMs for patients with PC that you could refer to instead of only referring to some of the existing PROMs (disclaimer: I am a co-author of one of these reviews).

A potential selection bias should be discussed, characterizing the group of patients who have used the tool (and have provided their contact details) and thereby qualified for participation, and how they might differ from the overall population of patients with PC.

"For the same reason, transcripts were not returned to the patients because this would be too burdensome" – The transcripts could have been provided to patients, who could then decide whether to read them and provide feedback, or whether this would be too burdensome.

COREQ checklist:

Some of the information required according to the COREQ checklist seems to be missing:

#5 "What experience or training did the researcher have?" It is only reported that one researcher "was trained in qualitative research".

I did not find information on the following three aspects in the manuscript:

#7 "What did the participants know about the researcher? e.g. personal goals, reasons for doing the research"

#8 "What characteristics were reported about the interviewer/facilitator? e.g. Bias, assumptions, reasons and interests in the research topic"

#25 "Description of the coding tree"

Minor / linguistic notes:

The phrase "reducing healthcare services" seems to be incomplete – did you mean reducing the use of, or the need for, healthcare services?

Please check the English translation of the interview guide ("Who and how where you diagnosed with post-COVID").

Is the English translation of the ABCC-tool the official, ready-to-use translation? According to PROM translation standards, this would require double translation and back translation. For example, the wording of item 4 ("did you suffer adjusting your life (e.g. planning of activities, get enough exercise, eating healthy)?") seems incorrect to me and should probably read something like "Did you have difficulty adjusting your lifestyle ...". While it is acceptable to provide an ad hoc translation, it should be noted that it is not ready for use with English-speaking patients.

One item (#6) seems to cover two very different symptoms ("did you suffer from pain or difficulties with breathing?"); maybe this is due to the translation only and what was meant is pain from breathing and difficulties with breathing?

"The grant of Pfizer was received by AHMGS. More information is available on <https://www.pfizer.nl/>." – Could you please where on the website the information can be found?

VERSION 1 - AUTHOR RESPONSE

Reviewer 1, Dr. GIACOMO GUIDO, University of Bari Aldo Moro

Thank you for your thorough comments.

Comments to the Author:

1. The study acknowledges that participants predominantly came from medium to high educational backgrounds, with no representation of individuals from low educational groups. This limitation deserves deeper reflection because educational level can directly influence health literacy, digital literacy, and therefore usability of patient-reported outcome tools. Patients with lower education or lower health literacy may find the ABCC-tool more difficult to navigate, potentially reducing its real-world applicability. I would suggest adding a section that explicitly discusses how this exclusion may have shaped the findings and what steps could be taken in future studies to ensure more equitable inclusion (for example, targeted recruitment or oversampling strategies for underrepresented groups).

Reply: We agree that this is a limitation that needs deeper consideration.

Change in article: We elaborated the paragraph in the limitation section.

Line 393-398: *Previous studies on the ABCC tool's content validity and patient experiences included participants with both higher and lower educational levels (10, 12). None of the patients reported difficulties with using the tool or understanding its language. Although the current study focused on a different patient population, the tool's purpose, functioning, and linguistic level remain unchanged. For future research, we*

recommend collecting basic participant information to enable more targeted recruitment and selection.

2. At the same time, and this is connected to the previous comment in a way, the extremely high attrition rate is concerning: out of more than 5,000 invited patients, only 19 interviews were completed. Even if this was due to technical, logistic, or patient-related reasons, such a discrepancy raises questions about the representativeness of the final sample. It would strengthen the paper to discuss potential reasons for this attrition (e.g., lack of interest, digital barriers, survey fatigue, recruitment strategy) and to consider how these factors might limit external validity. For example, patients who declined participation may systematically differ from those who took part (e.g., in age, severity of post-COVID symptoms, or socio-economic status).

Reply: we agree that this might lead to a potential selection bias.

Change in article: we added a paragraph to explain that this can limit the external validity of the results.

Line 382-387: *In total, only 143 of the 5615 invited patients were eligible and interested in participating. We suspect that concentration problems commonly associated with post-COVID may have discouraged many patients from completing the interviews, as these were perceived as too demanding, amongst others. This may have led to selection bias: those who volunteer to participate in scientific research are mostly woman, higher educated, and score higher on cognitive tests (1). The underrepresentation of patients with a low educational level could potentially limit the external validity of the findings.*

3. Some participants misunderstood the purpose of the tool, believing it to be related to treatment trials or clinical interventions rather than a self-assessment instrument. This is a crucial finding because it suggests that communication around PROMs can strongly influence patient engagement and expectations. The discussion would benefit from an expanded reflection on how such misperceptions can be prevented

Reply: thank you for suggesting this part for the discussion.

Change in article: we added a paragraph to the discussion.

Line 310-315: *For a few patients, the aim of the tool, and this study, was unclear and thought that the tool was part of a larger study to find a therapy or cure for post-COVID. This highlights how easily PROMs can be misunderstood, with potential consequences for engagement and expectations. To prevent such misperceptions, it is important that future use of the ABCC-tool is accompanied by clear communication about its purpose as a self-management and shared decision-making instrument, rather than as a diagnostic or therapeutic tool.*

4. Although the funding sources (Pfizer, ZonMw) are declared, the manuscript could be strengthened by clarifying the extent of funders' involvement in the study. For example, readers should be reassured that funders were not involved in study design, data collection, analysis, interpretation, or manuscript writing. Providing this explicit reassurance would help mitigate any concerns about potential conflicts of interest, particularly given that tool validation studies may lead to broader adoption and scaling

Reply: thank you for this suggestion

Change in article: we added that the funders had no role in the study design, execution of the study, or writing process.

Line 117-118: *The study was funded by Pfizer and ZonMw, however, they did not have any involvement in the study design, data collection, analysis or manuscript writing process.*

5. Several patient quotations are very long, which affects readability. Consider condensing them while retaining their meaning.

Reply: We have critically looked at the quotations, to assess their readability and the possibility of shortening them without losing their meaning.

Change in article: One quotation was shortened (line 270-274).

6. “ENGLISCH VERSION” in the appendix should be corrected to “ENGLISH VERSION.”

Reply: Thank you for noticing this typo.

Change in article: We corrected this in the manuscript (appendix 1)

7. Ensure uniform use of terms such as ABCC-tool, PROM, GP, and CXR throughout the manuscript. Currently, some are written with hyphens and others without

Reply: We carefully looked at the manuscript to make sure the terms were used uniformly.

8. In the conclusion, “balloon scheme” is used, while earlier sections use “balloon diagram”, better to harmonize

Reply: We agreed that this is confusing.

Change in article: we changed scheme to diagram.

Line 449-450: The most appreciated aspects of the tool are the balloon *diagram*, which presents the current and past health situation of the patient, and the diverse range of questions in the questionnaire.

9. The paper briefly notes directions for future research, but this could be expanded. For example, it would be valuable to suggest mixed-methods studies that include both patient and clinician perspectives to capture usability from both sides of the consultation. Plus, could it be implemented in other health systems besides the Dutch one? I would explore transferability.

Reply: Thank you for suggesting more ideas for future research.

Change in article: We have added more context to this paragraph.

Line 425-427: This study should include interviews with patients and healthcare professionals to evaluate the barriers and facilitators for using the tool and *study the usability from different perspectives*.

Line 437-440: *Although the ABCC-tool was originally developed in the Netherlands, future research could explore its implementation in other countries. This would require careful translation and cultural validation to ensure that the tool is both linguistically accurate and contextually appropriate for use in different healthcare settings. The first interest in the ABCC-tool of other countries has been reported already.*

10. The introduction provides a good overview of PROMs in post-COVID, but it could be strengthened by acknowledging that post-COVID outcomes extend beyond symptom monitoring to include functional and socio-economic consequences, such as impaired working ability. Recent studies have documented long-term work impairment up to two years after COVID-19 hospitalization (<https://doi.org/10.3390/v16050688>), which underscores the importance of PROMs that can also capture the impact of post-COVID on patients’ ability to participate in daily and professional life

Reply: Work impairment and (societal) participation is indeed an important burden for post-COVID patients.

Change in article: We have added this in the introduction.

Line 58-60: Over 200 different symptoms have been reported, manifesting in various combinations. These may include, but are not limited to, respiratory, cognitive, and neurological symptoms, (chronic) fatigue, emotional burden, *and work impairment(2-4)*.

Thank you for your comment.

Comments to the Author:

1. Overall: The manuscript is qualitative review of user feedback rather than a study. The article itself lacked clarity of what exactly is being assessed and the justification for doing so. Substantial revision is needed to better frame the contribution of this work.

The tool being assessed does not appear to be very useful, and the goal of evaluating the tool is not fully justified.

Reply: We thank the reviewer to point out that the article lacked clarity. We have added several points in the manuscript to be more comprehensive.

Regarding the statement that “the tool being assessed does not appear to be very useful”, we would kindly appreciate some clarification on what specific aspects led to this conclusion. We believe this is extensively discussed (especially in lines 302-309). Understanding the reviewer’s perspective would help us to better address this point and strengthen the justification and framing of our study in the revised version. We would also like to note that this concern was not raised by the other reviewers, which makes it difficult for us to understand why the tool is considered not very useful.

Change in article:

- We clarified the aim of the ABCC-tool and the purpose of this study
-

Line 41: The ABCC-tool is a promising instrument for post-COVID patients, offering a structured way to monitor and communicate *experienced burden* in addition to standard healthcare consultations.

Line 85-86: The ABCC-tool was chosen *because it measures burden of disease on a physical, emotional and social level*, and because of its person-centered approach and stimulation of self-management and shared decision making.

Line 102-106: In this study, we aimed to assess the content validity of the ABCC-tool for post-COVID and to evaluate its usability from the patients’ perspectives. *To the best of our knowledge, no other tool has the capability to measure physical, emotional, and social disease burden of post-COVID with the possibility to measure these disease burden in other chronic conditions, and visualize the results. Such tools are highly demanded by patients. Therefore, this evaluation was to ensure that the tool adequately covers all relevant content and to identify potential areas for improvement to make it easier for patients to use.*

- To clarify where in the manuscript is written that the tool appears useful, we added a little sentence to the discussion.
-

Line 302-309: Findings from this study confirm that the tool includes the most commonly reported post-COVID burdens and offers patients the opportunity to document additional concerns through an open-text feature. Furthermore, patients appreciated the tool for multiple reasons, *meaning that the tool appears useful*. First, the diversity of topics included in the tool, particularly as many of these are not routinely addressed in their healthcare. Second, patients found the tool's language accessible, because of its short sentences and simple wording (23). Third, the length of the questionnaire was acceptable. Fourth, the visualization gave an insightful overview of disease burden. Finally, the tool was easy to find and use online.

Thank you for your thorough comments.

Comments to the Author:

Abstract:

1. If possible given the limited word count, it might be helpful to specify the context in which the patients completed the tool already in the abstract.

Reply: We agree that including this information would benefit the abstract.

Change in article: We added this to the abstract while still meeting the word limit.

Line 28-29: This study explores the patients' perspective on the content of the ABCC-tool for post-COVID, and the tool's usability *in a home-based setting*.

Introduction:

2. There is contradictory information about the exact construct to be measured. For example, "offering a structured way to monitor and communicate symptoms" seems to indicate that the construct to be measured is PC symptoms, while elsewhere the construct is referred to as "physical, emotional, and social burden of disease".

Reply: The ABCC-tool measures burden of disease on physical, emotional and social level, and offers a structured way to identify the most important symptoms.

Change in article: We added more details about the ABCC-tool's purpose, so it is more clear that the tool has several aspects. Furthermore, to clarify, we changed symptoms to experienced burden.

Line 41: The ABCC-tool is a promising instrument for post-COVID patients, offering a structured way to monitor and communicate *experienced burden* in addition to standard healthcare consultations.

Line 85-86: The ABCC-tool was chosen *because it measures burden of disease on a physical, emotional and social level*, and because of its person-centered approach and stimulation of self-management and shared decision making.

3. I believe that the figure of "ten to twenty percent experience persistent symptoms that hinder their daily life activities and decrease quality of life" is probably outdated and was only true only in the early years of the pandemic. I suggest referencing more recent studies on incidence and/or prevalence.

Reply: We agree that these numbers were outdated. We changed these into more recent numbers reported by the WHO in 2025.

Change in article: We changed 10-20% to 6%.

Line 56-57: However, *according to the World Health Organization (WHO), six percent* experience persistent symptoms that hinder their daily life activities, and decrease quality of life (QoL) (2, 3, 6).

4. The considerable overlap between PC and ME/CFS could be mentioned, as could the phenomenon of post-exertional malaise (PEM), which is reported even more frequently.

Reply: We thank the reviewer for this valuable suggestion. We agree that there are notable similarities reported between Post-COVID condition and ME/CFS. However, a detailed comparison between the two conditions falls outside the scope of the present study.

We acknowledge that the perceived overlap between Post-COVID and ME/CFS has been emphasized especially within patient communities, who often seek to align their experiences and strengthen advocacy efforts. Therefore, we recommended to explore the potential of using the ABCC-tool for post-COVID for ME/CFS patients.

Change in article: we added as recommendation for future research to explore the possibility of using/adapting the ABCC-tool for post-COVID to ME/CFS.

Line 441-445: *Furthermore, it may be valuable to explore the potential application of the ABCC tool for post-COVID in patients with myalgic encephalomyelitis/chronic fatigue syndrome (ME/CFS), as the burden associated with ME/CFS closely resembles that of post-COVID. Both conditions are characterized by substantial health, economic, and social impacts, which are often underestimated due to limited awareness and understanding of these diseases (7).*

Methods:

5. Patients obviously provided an email address when completing the tool online. Was this mandatory? Did they agree to be contacted for study invitation?

Reply: Patients were approached via CuraVista, the platform through which they accessed the ABCC-tool, using newsletter-like communication. Patients were invited to contact the research team themselves when they were interested in the study. Only those who expressed interest in receiving further information actively contacted the research team. Importantly, patients had not provided explicit prior consent to be approached for research purposes, as this was not requested from them.

Change in article: We added clarification that the initial approach to patients was made through CuraVista.

Line 117-118: All patients who had used the ABCC-tool online via CuraVista (www.curavista.health) platform were invited for an online interview via email *by CuraVista*.

6. As the COSMIN guidelines provide recommendations on how to conduct content validation studies, you may wish to refer to these and which of these recommendations have been met in your work.

Reply: We have used the COSMIN methodology for assessing the content validity of PROMs.

Change in article: We have added this information to the manuscript.

Line 112-113: *The COSMIN methodology for assessing the content validity of PROMs was used to assess the content validity (8).*

7. "This qualitative study used an interpretative phenomenological approach"; elsewhere, you state that it used a thematic approach. I am not sure whether this work actually qualifies as interpretative phenomenological; from the methods and results, it seems to me more like a thematic analysis / content analysis.

Reply: a phenomenological approach focuses on understanding individuals' subjective experiences and perceptions of a specific phenomenon. The emphasis is on the essence of the participants' experiences. In this study, we wanted to explore the experiences of the patients with the tool's content and usability. The thematic approach mentioned in the manuscript refers to the coding process and focuses on identifying recurring themes or patterns in the data. It is flexible and often used to extract deeper meaning from qualitative data. With all respect, we therefore interpreted this design to be a suitable study design and analysis approach.

8. Do I understand correctly that the ABCC-tool, which had been developed for use in other chronic diseases, has not been adapted for use in patients with PC, meaning that this is the first study to investigate whether its content is appropriate to PC? I wondered because the item "did you suffer from a strange taste or smell, or tasted/smelled nothing" seems to be quite PC-specific. On the other hand, it is usually recommended that the development of a tool is performed with patient input (via qualitative data). This may explain why some potentially quite relevant impairments are not covered, such as PEM (while feeling worse after exercising is covered, feeling worse after any kind of activity is not).

Reply: the ABCC-tool has been developed for several other chronic conditions (e.g. COPD, diabetes, and osteoarthritis). For all conditions, patients complete a module with generic questions (with topics like fatigue, difficulties in daily activities) and lifestyle module (alcohol use, smoking, etc.). Then, patients get one or more disease specific modules. For post-COVID patients, this is the post-COVID module.

This is indeed the first study on the post-COVID module, however, a lot of studies (9-21) have been done for the other chronic conditions. To develop the ABCC-tool for post-COVID, we included both literature and input from patients/healthcare providers to develop the post-COVID module.

Furthermore, regarding PEM, the reviewer is referring to question 27 (if you felt worse after exercising, how long did this last?). This question focuses on the time of feeling worse. Questions 29-31 refer to being limited after strenuous, moderate, and daily activities. According to the patients in the interviews, these questions were sufficient to address these burdens.

9. It is all the more important to thoroughly test the appropriateness and completeness of the tool with this new patient group. However, if I understand correctly, the tool was not presented again in the interviews; instead, participants had to rely on their recollection of the items. This severely limits the conclusions that can be drawn regarding comprehensibility and in particular completeness of the tool's content, even more so as multiple patients had to be excluded as they did not remember using the tool at all. The COSMIN guidelines, for example, request that patients should be asked about each item separately, which has not been done here.

Reply: during the interviews, patients had a copy of the questionnaire to recall the questions. Reading the comment of Reviewer 3, we understand that this was not clear enough. Furthermore, relevancy and comprehensibility were asked per question.

Change in article: We added that patients receive the tool by mail to minimize recall bias. Also, we clarified the topic list that relevancy and comprehensibility was asked per item in the questionnaire during the topic list.

Line 140-141: Prior to the interviews, patients received a printed copy of the tool's questionnaire and the balloon diagram by mail *to recall the tool and* facilitate discussion

Appendix 2:

16. Was question ... relevant for you? (*per item in the questionnaire*)

17. Was question ... formulated clearly? (*per item in the questionnaire*)

10. According to the interview guide, patients were asked whether they thought the ABCC tool was complete. However, they were not provided with any information about what the instrument intends to measure. What exactly does "complete" refer to in this context?

Reply: The word 'complete' lost a little context in the translation of the Dutch word 'compleet' to English.

Change in article: We added the word: 'comprehensive' (which is a better translation) to the topic list to ensure the meaning of the question.

Appendix 2: 15. Was the ABCC-tool complete (*comprehensive*)?

11. Were the computer-assisted transcriptions checked for correctness?

Reply: Yes, they were all checked for correctness.

Change in article: We added this information to the method section.

Line 150: Transcripts were 100% checked for correctness.

12. How is lifestyle transformed into balloon colour? For example, "moderately intense physical exercise for 30 minutes" (item 36) may be considered an unhealthy lifestyle choice for many people with PEM; therefore, a negative response should not be translated into a red balloon.

Reply: This is a very relevant and interesting remark. The lifestyle balloon is now coded based on the current recommendation regarding the different lifestyle factors. So, this indeed means that a red balloon will pop up when someone is not physically active for 30 minutes a day. This point therefore needs to be addressed in the discussion.

Change in article: We added that these balloon scores are not always appropriate for everyone, and therefore needs to be evaluated again.

Line 418-423: A few adjustments (open text fields, periodic reminders, including a figure legend, and clarifying the purpose of the tool) need to be made to the tool, based on the findings in this research. *Furthermore, although the questions of the tool have now been validated, a thorough evaluation of the other aspects of the tool is in place. For example, the recommended score calculation and lifestyle advice regarding physical activity might not be appropriate in the case of post-exertion malaise (PEM). This needs further investigation.*

Results:

13. How did you assess which other chronic conditions the participants had?

Reply: This was asked during the interviews. Question 2 regards the patients' health status, and in case the patient did not tell anything about other chronic conditions, this was asked as follow-up question. However, we agree that this might not be clear, and therefore, we added this sub-question, in line with the followed procedure.

Change in article:

Appendix 2: 2a. *Do you have other chronic conditions?*

14. "One interview had to be cut short on the demand of the patient. Additionally, some patients mentioned at the beginning of the interview not to be able to talk longer than a specific time. In that case, the interviewer started with the most important topics of the study". How many patients exactly had limited or missing data compared to the interview guide? This information could be added to the patient characteristics table.

Reply: Only 1 interview was cut short early.

Change in article: We added this information to the result section.

Line 171-172: *One interview (interview 11) was cut short early due to concentration issues.*

15. You state that data saturation was not achieved for comprehensiveness but chose not to include additional patients. In the discussion, you argue that "Due to the complexity of post-COVID and the aforementioned over 200 symptoms that are associated with this condition, it is not feasible to include all symptoms in the questionnaire. We therefore decided that this issue can only be solved by incorporating an open question" – I believe that this may be insufficient. More information is needed on how data saturation has been assessed and on the empirical basis for the statement that "the most prevalent burdens are included into the tool".

Reply: We agree that the information was missing on how we determined data saturation.

Change in article: We added that new symptoms and themes were monitored during the interviews, in the method-, result-, and discussion section.

Line 153-154: *New themes were systematically monitored in between interviews to observe data saturation and information power.*

Line 239-241: Regarding the comprehensiveness of the tool's questionnaire, it was challenging to reach data saturation due to the complexity of the disease and the variety of ways in which the burden experienced manifests itself. *However, patients agreed that the most burdensome and most frequent burdens are included into the questionnaire.*

Line 367-381: Data saturation was not fully achieved on comprehensiveness, *however, we systematically monitored whether new symptoms or themes emerged during the interviews. While some additional complaints were mentioned, these largely overlapped with issues already identified, or were not applicable for the ABCC-tool. We therefore argue that information power was sufficient, as the sample provided rich and relevant data on the clarity and relevance of the questionnaire items.* Factors such as the study's aim and the quality of the collected data play an important role in determining whether the sample holds sufficient information for meaningful analysis (5). Due to the complexity of post-COVID and the aforementioned over 200 symptoms that are associated with this condition (3), it is not feasible to include all symptoms in the questionnaire. *Furthermore, the included symptoms reflect the most prevalent burdens described in the literature on post-COVID, such as fatigue and cognitive dysfunction, which were consistently confirmed by our participants (2-4).* We therefore decided that this issue can only be solved by incorporating an open question asking complaints other than those already mentioned. This way, the most prevalent burdens are included into the tool, with space for additional burdens to report.

16. I also have some concerns about the heterogeneity of the sample, and therefore the generalisability of the results to the overall population of people with PC. Out of 5615 patients invited for an interview, only 143 responded, which seems quite low. The youngest person included was 41, so the perspectives of younger patients are not represented. Also, severely affected participants, e.g. those with severe ME/CFS, were not (and probably could not be) included, which should be discussed.

Reply: Thank you for pointing this out. We agree that our sample was not heterogenous in all possible ways, for example age or severity of symptoms. We only filtered patients on whether they could remember the ABCC-tool and balloon diagram.

Specifically for patients with severe symptoms, we speculate that the interview could be too burdensome, and therefore these individuals may have been less interested in participating.

Change in article: We elaborated in the limitation section that there is a potential selection bias.

Line 382-387: *In total, only 143 of the 5615 invited patients were eligible and interested in participating. We suspect that concentration problems commonly associated with post-COVID may have discouraged many patients from completing the interviews, as these were perceived as too demanding, amongst others. This may have led to selection bias: those who volunteer to participate in scientific research are mostly woman, higher educated, and score higher on cognitive tests (1). The underrepresentation of patients with a low educational level could potentially limit the validity of the findings.*

17. Many issues with the tool were found during the interviews, but these did not lead to revisions of the tool (except for the addition of a legend). Examples of issues that were not addressed include the recall period, the purpose of the tool and how to interpret the graphic. Instead, it is recommended that only "Minor adjustments, e.g. open text fields and periodic reminders, need to be made to the tool, based on the findings in this research" (which should be specified: which adjustments exactly should be made, as only examples are provided?). It seems to me that the findings do not fully justify the conclusion that the tool showed content validity and feasibility.

Reply: Thank you for pointing this out. The recall period was only for purpose of this study, and therefore not necessary to revise. However, it is indeed important to clarify the purpose of the questionnaire before completion.

Change in article: we added more context to what these adjustments should be.

Line 418-420: *A few adjustments (open text fields, periodic reminders, including a figure legend, and clarifying the purpose of the tool) need to be made to the tool, based on the findings in this research.*

Discussion:

18. The need for this new tool and why it is innovative as compared to existing tools are discussed. Note that there are reviews of existing PROMs for patients with PC that you could refer to instead of only referring to some of the existing PROMs (disclaimer: I am a co-author of one of these reviews).

Reply: Thank you for mentioning the review.

Change in article: We have added the review to the manuscript.

Line 316-320: *Since the emergence of post-COVID, several PROMs have been developed and validated to assess the disease burden, on functional status, symptom burden, QoL, and post-COVID related stigma (22). Examples of tools are the Symptom Burden Questionnaire for Long COVID (SBQ-LB), the Post COVID-19 Condition Stigma Questionnaire (PCCSQ), and the Long COVID Symptoms and Severity Score (LC-SSS scale) (23-26).*

19. A potential selection bias should be discussed, characterizing the group of patients who have used the tool (and have provided their contact details) and thereby qualified for participation, and how they might differ from the overall population of patients with PC.

Reply: Thank you for noticing this. We agree that there is a potential selection bias.

Change in article: We added this to the discussion.

Line 382-387: *In total, only 143 of the 5615 invited patients were eligible and interested in participating. We suspect that concentration problems commonly associated with post-COVID may have discouraged many patients from completing the interviews, as these were perceived as too demanding, amongst others. This may have led to selection bias: those who volunteer to participate in scientific research are mostly woman, higher educated, and score higher on cognitive tests (1). The underrepresentation of patients with a low educational level could potentially limit the validity of the findings.*

20. "For the same reason, transcripts were not returned to the patients because this would be too burdensome" – The transcripts could have been provided to patients, who could then decide whether to read them and provide feedback, or whether this would be too burdensome.

Reply: Thank you for pointing this out. We agree that this could have been a more suitable strategy; however, it was not applied in the present study. We will take this into account in future research.

Change in article: We added that giving the participants the opportunity to review their transcripts would have provided additional feedback.

Line 412-415: *For the same reason, transcripts were not returned to the patients because this would be too burdensome. However, we acknowledge that offering participants the option to review their transcripts could have provided an additional opportunity for validation and feedback, and we will consider this approach in future studies.*

COREQ checklist:

21. Some of the information required according to the COREQ checklist seems to be missing:

- #5 "What experience or training did the researcher have?" It is only reported that one researcher "was trained in qualitative research".

Reply: thank you for pointing out that we lacked information about the other researchers.

Change in article: We added the lacking information.

Line 135: *All other researchers involved had experience in performing qualitative research.*

22. I did not find information on the following three aspects in the manuscript:

- #7 "What did the participants know about the researcher? e.g. personal goals, reasons for doing the research"

Reply: We agree that this information is limited.

Change in article: We added that patients only knew the research aim.

Line 143: *Patients only were familiar with the aim of the research.*

- #8 "What characteristics were reported about the interviewer/facilitator? e.g. Bias, assumptions, reasons and interests in the research topic"

Reply: Thank you for pointing this out.

Change in article: There were no characteristics reported about the interviewer. We believe this is already added for #7 of the COREQ. (The interviewer had no relationship with any of the patients. Patients only were familiar with the aim of the research.) we added that the researcher was introduced as a health scientist and researcher at Maastricht University.

Line 143-145: The interviewer was introduced to the participants as a health scientist and researcher at Maastricht University.

- #25 "Description of the coding tree"

Reply: We believe this item of the COREQ is pointed out in line (Related codes were grouped into broader categories, and overarching themes were synthesized to gain a complete understanding of the patients' perspectives.)

Minor / linguistical notes:

23. The phrase "reducing healthcare services" seems to be incomplete – did you mean reducing the use of, or the need for, healthcare services?

Reply: this was indeed unclear.

Change in article: We specified that it was about the use of healthcare services.

Line 87-88: These are important factors for disease coping, improving health outcomes, and reducing *the use of* healthcare services

24. Please check the English translation of the interview guide ("Who and how where you diagnosed with post-COVID").

Reply: thank you for noticing this error.

Change in article: we corrected the translation.

Appendix 2: By whom and how where you diagnosed with post-COVID

25. Is the English translation of the ABCC-tool the official, ready-to-use translation? According to PROM translation standards, this would require double translation and back translation. For example, the wording of item 4 ("did you suffer adjusting your life (e.g. planning of activities, get enough exercise, eating healthy)?") seems incorrect to me and should probably read something like "Did you have difficulty adjusting your lifestyle ...". While it is acceptable to provide an ad hoc translation, it should be noted that it is not ready for use with English-speaking patients.

Reply: This translation is not validated, and only for the purpose of this article.

Change in article: we added a sentence to make this statement

Appendix 1: (Note: this is an unvalidated translation from Dutch to English specifically for this paper)

26. One item (#6) seems to cover two very different symptoms ("did you suffer from pain or difficulties with breathing?"); maybe this is due to the translation only and what was meant is pain from breathing and difficulties with breathing?

Reply: We agree that this can be confusing. We meant: pain while breathing or difficulties with breathing.

Change in article: To clarify this, we changed the 'or' to '/'.

*Appendix 1: did you suffer from *pain/ difficulties* with breathing?*

27. "The grant of Pfizer was received by AHMGS. More information is available on <https://www.pfizer.nl/>."
– Could you please where on the website the information can be found?

Reply: This website was consulted as a general reference to provide contextual information on Pfizer as a company.

Other remarks

We noticed a minor error in appendix 1: 1 question was missing in the appendix of the manuscript, and 1 question was presented as 1 question which should be 2 questions. Therefore, the total number of questions in the questionnaire was added with 2. This does not change anything for the results of the present study.

1. Tripepi G, Jager KJ, Dekker FW, Zoccali C. Selection bias and information bias in clinical research. *Nephron Clinical Practice*. 2010;115(2):c94-c9.
2. Cha C, Baek G. Symptoms and management of long COVID: A scoping review. *Journal of Clinical Nursing*. 2024;33(1):11-28.
3. World Health Organization. Post COVID-19 condition (Long COVID) 2022 [Available from: <https://www.who.int/europe/news-room/fact-sheets/item/post-covid-19-condition>].
4. Frallonardo L, Ritacco AI, Amendolara A, Cassano D, Manco Cesari G, Lugli A, et al. Long-Term Impairment of Working Ability in Subjects under 60 Years of Age Hospitalised for COVID-19 at 2 Years of Follow-Up: A Cross-Sectional Study. *Viruses*. 2024;16(5):688.
5. Malterud K, Siersma VD, Guassora AD. Sample size in qualitative interview studies: guided by information power. *Qualitative health research*. 2016;26(13):1753-60.
6. World Health Organization. Post COVID-19 condition (long COVID) 2025 [
7. Vester P, Boudouoglou-Walter S, Schreyögg J, Wieting C, Blome C. Burden of Disease in Myalgic Encephalomyelitis/Chronic Fatigue Syndrome (ME/CFS): A Scoping Review. *Appl Health Econ Health Policy*. 2025.
8. Terwee CB, Prinsen C, Chiarotto A, De Vet H, Bouter LM, Alonso J, et al. COSMIN methodology for assessing the content validity of PROMs—user manual. Amsterdam: VU University Medical Center. 2018.
9. Boudewijns EA, Claessens D, van Schayck OC, Keijsers LC, Salomé PL, Bilo HJ, et al. ABC-tool reinvented: development of a disease-specific 'Assessment of Burden of Chronic Conditions (ABCC)-tool' for multiple chronic conditions. *BMC family practice*. 2020;21(1):1-7.
10. Boudewijns EA, Claessens D, van Schayck OC, Twellaar M, Winkens B, Joore MA, et al. Effectiveness of the Assessment of Burden of Chronic Conditions (ABCC)-tool in patients with asthma, COPD, type 2 diabetes mellitus, and heart failure: A pragmatic clustered quasi-experimental study in the Netherlands. *European Journal of General Practice*. 2024;30(1):2343364.

11. Claessens D, Boudewijns EA, Keijsers LC, Gidding-Slok AH, Winkens B, van Schayck OC. Validity and Reliability of the Assessment of Burden of Chronic Conditions Scale in the Netherlands. *The Annals of Family Medicine*. 2023;21(2):103-11.
12. Claessens D, Boudewijns EA, Vervloet M, Keijsers LC, Gidding-Slok AH, van Schayck OC, et al. Barriers and facilitators to the implementation of the Assessment of Burden of Chronic Conditions tool in Dutch primary care: a context analysis. *BMJ open*. 2025;15(1):e087197.
13. Claessens D, Vervloet M, Boudewijns EA, Keijsers L, Gidding-Slok AHM, van Schayck OCP, et al. Understanding the healthcare providers' perspective for bringing the assessment of burden of chronic conditions tool to practice: a protocol for an implementation study. *BMJ Open*. 2023;13(3):e068603.
14. Claessens D, Vervloet M, Boudewijns EA, Keijsers LC, Gidding-Slok AH, van Schayck OC, et al. Process evaluation of the implementation of the assessment of burden of chronic conditions tool in Dutch primary care—lessons from a qualitative implementation study. *BMC Health Services Research*. 2024;24(1):827.
15. Debie VHJ, Boymans TAEJ, Gidding-Slok AHM, van Schayck OCP, Ottenheijm RPG. The Assessment of Burden of Chronic Conditions (ABCC-) Tool: A Valid and Reliable Tool for Hip, Knee, Hand, Wrist, Foot and Ankle Osteoarthritis. *Osteoarthritis and Cartilage Open*. 2025;7(3):100623.
16. Debie VHJ, Boymans TAEJ, Ottenheijm RPG, van Schayck OCP, Gidding-Slok AHM. Expanding the ABCC-tool for Osteoarthritis: Development and Content Validation. *Osteoarthritis and Cartilage Open*. 2024;6(3):100488.
17. Peters LHL, Joore MA, Gidding-Slok AHM, Keijsers LCEM, Twellaar M, Boudewijns EA, et al. Cost-effectiveness analysis of the Assessment of Burden of Chronic Conditions (ABCC) tool in primary care in the Netherlands *BMJ Open*. 2025;15(6):e099762.
18. Slok AH, Kotz D, van Breukelen G, Chavannes NH, Rutten-van Mólken MP, Kerstjens HA, et al. Effectiveness of the Assessment of Burden of COPD (ABC) tool on health-related quality of life in patients with COPD: a cluster randomised controlled trial in primary and hospital care. *BMJ open*. 2016;6(7):e011519.
19. Slok AH, Twellaar M, Jutbo L, Kotz D, Chavannes NH, Holverda S, et al. 'To use or not to use': a qualitative study to evaluate experiences of healthcare providers and patients with the assessment of burden of COPD (ABC) tool. *npj Primary Care Respiratory Medicine*. 2016;26(1):1-8.
20. Slok AHM, Bemelmans TCH, Kotz D, van der Molen T, Kerstjens HAM, in 't Veen JCCM, et al. The Assessment of Burden of COPD (ABC) scale: a reliable and valid questionnaire. *COPD: Journal of Chronic Obstructive Pulmonary Disease*. 2016;13(4):431-8.
21. Slok AHM, Chavannes NH, van der Molen T, Rutten-van Mólken MPMH, Kerstjens HAM, Salomé PL, et al. Development of the Assessment of Burden of COPD tool: an integrated tool to measure the burden of COPD. *NPJ Primary Care Respiratory Medicine*. 2014;24(1):1-4.
22. Baalman A-K, Blome C, Stoletzki N, Donhauser T, Apfelbacher C, Piontek K. Patient-reported outcome measures for post-COVID-19 condition: a systematic review of instruments and measurement properties. *BMJ Open*. 2024;14(12):e084202.
23. Hughes SE, Haroon S, Subramanian A, McMullan C, Aiyegbusi OL, Turner GM, et al. Development and validation of the symptom burden questionnaire for long covid (SBQ-LC): Rasch analysis. *Bmj*. 2022;377:e070230.
24. Damant RW, Rourke L, Cui Y, Lam GY, Smith MP, Fuhr DP, et al. Reliability and validity of the post COVID-19 condition stigma questionnaire: A prospective cohort study. *EClinicalMedicine*. 2023;55:101755.
25. Ye G, Zhu Y, Bao W, Zhou H, Lai J, Zhang Y, et al. The Long COVID Symptoms and Severity Score: Development, Validation, and Application. *Value Health*. 2024;27(8):1085-91.
26. Tran VT, Riveros C, Cleprier B, Desvarieux M, Collet C, Yordanov Y, et al. Development and Validation of the Long Coronavirus Disease (COVID) Symptom and Impact Tools: A Set of Patient-Reported Instruments Constructed From Patients' Lived Experience. *Clin Infect Dis*. 2022;74(2):278-87.

VERSION 2 - REVIEW

Name **Blome, Christine**
Affiliation **University Medical Center Hamburg-Eppendorf , Health Services Research in Dermatology**
Date **06-Nov-2025**
COI

Thank you very for considering my suggestions and for your helpful explanations. A few questions remained on my part:

Ad your reply to comment 4: A substantial number of patients with Post-Covid do not only have symptoms that resemble those of ME/CFS but actually *have* ME/CFS (with SARS-CoV-2 as a new viral trigger for the disease in addition to previously identified triggers such as EBV or influenza, meaning that in many cases Post-Covid is ME/CFS). Therefore, I believe it should at least be mentioned that many people with Post-Covid (with estimates ranging from 20-50% to my knowledge) fulfil the diagnostic criteria for ME/CFS.

Ad your reply to comment 8: "Then, patients get one or more disease specific modules. For post-COVID patients, this is the post-COVID module." Do I understand correctly that some Post-COVID-specific items have been added to the ABCC tool prior to its evaluation in this study? I could not find this information in the manuscript. I believe this information should be included, along with details of how these additional items were developed.

Ad your reply to comment 10: What I meant to say was that completeness/comprehensiveness can only be evaluated when patients know what the tool is meant to cover (here, the burden of disease) – that is, complete with regard to what? This information has obviously not been provided to participants. However, as the data has already been collected and participants obviously did have some idea about the construct, I do not suggest making any additional changes to the text.

VERSION 2 - AUTHOR RESPONSE

Reviewer 3 - Dr. Christine Blome, University Medical Center Hamburg-Eppendorf

1. Ad your reply to comment 4: A substantial number of patients with Post-Covid do not only have symptoms that resemble those of ME/CFS but actually *have* ME/CFS (with SARS-CoV-2 as a new viral trigger for the disease in addition to previously identified triggers such as EBV or influenza, meaning that in many cases Post-Covid is ME/CFS). Therefore, I believe it should at least be mentioned that many people with Post-Covid (with estimates ranging from 20-50% to my knowledge) fulfil the diagnostic criteria for ME/CFS.

Reply: This is indeed an important addition. Thank you for pointing this out.

Change in article: We added these percentages to the discussion.

Line 444-446: *In addition to the similarities in perceived disease burden, several studies have examined the prevalence of post-COVID and ME/CFS. These studies reported that 8–58% of participants met criteria for both post-COVID and ME/CFS [40-42].*

2. Ad your reply to comment 8: "Then, patients get one or more disease specific modules. For post-COVID patients, this is the post-COVID module." Do I understand correctly that some Post-COVID-specific items have been added to the ABCC tool prior to its evaluation in this study? I could not find this information in the manuscript. I believe this information should be included, along with details of how these additional items were developed.

Reply: The ABCC-tool is built in a modular format: every patient completes a generic questionnaire with questions that are applicable to (almost) everyone with a chronic condition (e.g. fatigue, difficulties with daily life activities, difficulties with social activities). Then, depending on what chronic condition a patient has, those disease-specific modules are added to the generic module. These disease specific modules are specific for a certain disease. So far, these are developed for: COPD, asthma, diabetes mellitus type 2, chronic heart failure, cardiovascular risk management, osteoarthritis, and post-COVID. This generic module was developed in a previous study, and these disease specific modules are added one by one. So, Reviewer 3 is correct that some Post-COVID-specific items have been added to the ABCC tool prior to its evaluation in this study.

Change in article: we added the modular format of the tool to the introduction

Line 84-88: *As the successor to the ABC-tool, the ABCC-tool extends its focus beyond chronic obstructive pulmonary disease (COPD). Because of its modular format, every patient completes a set of generic questions, followed by one or more disease-specific questions. Until now, these disease-specific questions has been developed for asthma, COPD, type 2 diabetes mellitus, chronic heart failure, osteoarthritis, cardiovascular risk management, and post-COVID [9-11].*

3. Ad your reply to comment 10: What I meant to say was that completeness/comprehensiveness can only be evaluated when patients know what the tool is meant to cover (here, the burden of disease) – that is, complete with regard to what? This information has obviously not been provided to participants. However, as the data has already been collected and participants obviously did have some idea about the construct, I do not suggest making any additional changes to the text.

Reply: Thank you for elaborating on your previous comment. We understand what you mean and agree with your suggestion. We therefore did not make any additional changes to the text.